# The Surface Properties of Implant Materials by Deposition of High-Entropy Alloys (HEAs)

**DOI:** 10.3390/nano13061123

**Published:** 2023-03-21

**Authors:** Khalid Usman, Doori Kang, Geonwoo Jeong, Khurshed Alam, Athira Raveendran, Jinhui Ser, Woohyung Jang, Hoonsung Cho

**Affiliations:** 1School of Materials Science & Engineering, Chonnam National University, Gwangju 61186, Republic of Korea; 2Department of Prosthodontics, School of Dentistry, Chonnam National University, Gwangju 61186, Republic of Korea

**Keywords:** high-entropy alloys (HEAs), sputtering, corrosion, implants, biomaterial

## Abstract

High-entropy alloys (HEAs) contain more than five alloying elements in a composition range of 5–35% and with slight atomic size variation. Recent narrative studies on HEA thin films and their synthesis through deposition techniques such as sputtering have highlighted the need for determining the corrosion behaviors of such alloys used as biomaterials, for example, in implants. Coatings composed of biocompatible elements such as titanium, cobalt, chrome, nickel, and molybdenum at the nominal composition of Co_30_Cr_20_Ni_20_Mo_20_Ti_10_ were synthesized by means of high-vacuum radiofrequency magnetron (HVRF) sputtering. In scanning electron microscopy (SEM) analysis, the coating samples deposited with higher ion densities were thicker than those deposited with lower ion densities (thin films). The X-ray diffraction (XRD) results of the thin films heat treated at higher temperatures, i.e., 600 and 800 °C, revealed a low degree of crystallinity. In thicker coatings and samples without heat treatment, the XRD peaks were amorphous. The samples coated at lower ion densities, i.e., 20 µAcm^−2^, and not subjected to heat treatment yielded superior results in terms of corrosion and biocompatibility among all the samples. Heat treatment at higher temperatures led to alloy oxidation, thus compromising the corrosion property of the deposited coatings.

## 1. Introduction

High-entropy alloys (HEAs) are emerging advanced materials, and they are also called multicomponent alloys, multi-principal-element alloys, and compositionally complex alloys (CCAs). These alloys were first reported in 2004 by two independent research groups, namely Yeh et al. [1] and Cantor et al. [2]. The alloy reported by the latter group was composed of an equimolar mixture of Cr, Mn, Fe, Co, and Ni (i.e., Cantor alloy [3]). Owing to the synergic property of HEAs, they appear to be a better alternative to existing biomaterials such as 316 L stainless steel, titanium, and its derivative alloys such as Ti 6Al 4V, as well as conventional alloys (CAs) such as ASTM F75, F90, and F562 (HS251-Haynes Stellite), also called Vitallium [4], in terms of corrosion and biocompatibility.

Over a broad temperature range, slow diffusion, lattice distortion, and high entropy are the defining characteristics of HEAs as compared to conventional alloys (CAs). The mixture of these properties affects the corrosion resistance, tribocorrosion, strength, hardness, ductility, wear, and erosion of HEAs [5]. HEAs have the potential for application in various industries, including aerospace, energy, refractory materials, and three-dimensional (3D) printing, in addition to their use as biomaterials in implants or surgical equipment.

The constituent materials of the Co_30_Cr_20_Ni_20_Mo_20_Ti_10_ HEA were selected from the perspective of biological compatibility. Co-Cr and its oxide states, e.g., Cr (VI), are considered CMR substances and cause lung cancer [6]. Ni sensitization is frequently accompanied by Co sensitization. There was no indication of Ti-specific hypersensitivity [7]. To ensure the neoformation of tissues and organs and the cells in a scaffold, the surface must be compatible anatomically, physiologically, and histologically. New bone formation can be reinforced by using bioactive signals such as genes, tethered agents, soluble factors, total growth factor (TGF-β), and its subgroups such as bone morphogenetic protein (BMP) [8,9]. Protein adhesion initiates the initial phases of corrosion in human implants. Proteins either adhere permanently to metallic implants or depart from them through flow assisted by human bodily fluids, carrying some metallic fragments in the process. The disadvantage of conventional Ti-metallic implants is that it has strength ductility mismatch with bones and also release extra low interstitial (ELI) of Ti-6Al-4V leading to phosphorus deficiency in blood and bones [10].

To synthesize the HEA, a few basic parameters must be considered, including the Hume-Rothery rule [11,12], intrinsic properties such as entropy and enthalpy of mixing [13], valence electron configuration (*VEC*) [14], and rules for single phase, crystalline, intermetallic materials [3,15,16]. HEAs can be designed by evaluating the properties of materials that constitute HEA [17,18]. Alternatively, they can be synthesized by using computing techniques such as the phase-diagram calculation [19]. The principal equations that govern the properties of HEAs are as follows:(1)$$\Delta S_{mix}=R\sum_{i}^{n}c_{i}\;ln\;c_{i}$$
(2)$$VEC=\sum_{i=1}^{n}c_{i}{\left(VEC\right)}_{i}$$
(3)$$\delta =100\%\sqrt{\sum_{i=1}^{n}c_{i}{\left(1-\frac{r_{i}}{\sum_{j=1}^{n}c_{j}r_{j}}\right)}^{2}}$$
(4)$$\Delta H_{mix}=\sum_{i=1,i\neq j}^{n}{Ω}_{ij}c_{i}c_{j}$$
(5)$$Ω=\frac{T_{m}\Delta S_{mix}}{\left|\Delta H_{mix}\right|}$$
where Δ*S_mix_* is the entropy of mixing, *R* is the universal gas constant, *VEC* denotes the valence electron configuration, delta (δ) is a unitless value indicating atomic size difference, Δ*H_mix_* is the enthalpy of mixing, Ω is a dimensionless parameter that denotes energy gain, *T_m_* is the melting point, and *c_i_* is the atomic composition. These mathematical expressions are used to evaluate the alloy systems.

In conventional methods such as casting and powder metallurgy, samples must be recast and pressed several times under vacuum to achieve homogeneity [20]. Moreover, even after these processes, the elements do not mix with complete miscibility and form amalgamations in certain microscopic regions. By contrast, sputtering, a type of physical vapor deposition (PVD) method, involves more spontaneous mixing of the constituent materials. Moreover, it does not require additional recasting and pressing steps. With sputtering, the layer thickness of HEA, which impacts the corrosion resistance and biocompatibility of the underlying substrate, can be controlled easily. In this work, corrosion analyses of sputtered HEA thick and thin films in simulated body fluid (SBF) are performed, whereas in existing studies, the corrosion properties of HEA were studied in NaCl, H_2_SO_4_, and seawater. During sputtering, the films are subjected to a metastable-phase energy barrier of E~0.25 eV before their final adhesion to the substrate surface, and this metastable phase is the transition region between the gasification and substrate phases [21].

## 2. Materials and Methods

The top-down configuration of an Artec system magnetron sputtering machine was used for HEA deposition. This machine is equipped with a target holder measuring 4.25 in. The target holder was upgraded to accommodate 20 pieces. In total, six cobalt, two titanium, and four nickel, chromium, and molybdenum were cut by means of electron discharge machining (EDM). Approximately 99.99% pure elemental discs were procured from THIFINE (the film and fine materials ©). To avoid charge accumulation at the target, the radiofrequency (RF) sputtering process was used, as illustrated in Figure 1a. The frequency ranged from 5 to 30 MHz, and the typical frequency was 13.56 MHz; electron densities ranged from 10^9^ to 10^11^ cm^−3^, and discharge pressure ranged from 0.5 to 10 mTorr [22]. Rare-earth magnets were used in the RF magnetron sputtering process. Figure 1b depicts how the magnetic field triggered a spiral momentum of the electrons around the target. This momentum increases the probability of sustained plasma, even at low pressures [23]. A 5 × 10^−2^ mbar vacuum was created before turning on the main valve (M/V). The samples were prepared at two different ion current densities of 2 × 10^−5^ A/cm^2^ and 5 × 10^−5^ A/cm^2^ [24]. Lower ion current densities reduce the likelihood of substrate oxidation [25]. After the thin HEA films were deposited by means of sputtering, the samples were post-treated in a multifunctional vacuum furnace (Ajeon Heating Industrial Co., Ltd., Namyangju-si, Republic of Korea) for 1 h at three different temperatures of 400 °C, 600 °C, and 800 °C. In this treatment, the maximum temperature was attained with a low heating curve rate, i.e., 5 °C per minute, to avoid large temperature gradients that result in tensile stresses and lead to the Kirkendall effect in coatings [26]. The treated samples were then cooled in a furnace under a 0.14 SCCM argon flow.

Thermodynamic calculations were performed using mathematical expressions. From Equation (3), the average atomic radius r_avg_ of the prepared HEA Co_30_Cr_20_Ni_20_Mo_20_Ti_10_ was calculated as 1.2931 Å. The atomic size difference was 9.4%. By applying Equation (2) to the XPS composition analysis result of the sample (Table 1), the nominal valence electron configuration (*VEC*) was calculated as 7.15. The enthalpy of mixing was calculated using Equation (4) (Δ*H_mix_* was taken as −4.07 KJ/mol) [27,28].

Corrosion experiments were conducted using BioLogic’s VSP 300 potentiostat and EC-Lab software (v11.33). For corrosion testing, the Metek-designed flat cell K0235 was used, which exposes 1 cm^2^ of the working electrode surface for greater accuracy. The cell was designed according to ASTM G5 with constant argon purging [29]. The fact that the oxygen in electrolytes reaches the surface of HEA affects the rate of oxidation on the HEA. By combining Fick’s law for diffusion and Faraday’s law, one can calculate the cathodic current associated with oxygen diffusion [30]. A simulated body fluid called Ringer’s solution was selected as the electrolyte for corrosion testing, and Bode plots were constructed to analyze the electrochemical impedance of the synthesized HEA [31,32,33,34].

Cellular metabolic activities were monitored by developing a 3-(4,5-dimethylthiazol-2-yl)-2,5-diphenyl-2H-tetrazolium bromide (MTT) assay of osteoblast cell type (3T3 cells). MC 3T3 (osteoblast cells) were taken from a cryogenic freezer and grown in an incubator for one month in regular alpha-minimum essential medium (MEM) without L-ascorbic acid and phenol red at 37 °C and with 5% CO_2_ supply. After the cells grew to 80% confluence in a T-75 flask, they were sufficiently populated. Twice each week, the medium was changed, and the MTT assay was developed when the cell confluence exceeded 1 × 10^5^ cells/mL [35].

### Thin-Film Characterization

X-ray diffraction analysis (XRD, using Cu-K radiation) was performed to examine the crystallinity and phase structure of the synthesized HEAs. A low scan rate (1°/min) was employed to obtain a higher resolution. The step size, 2θ, and omega range of the goniometer were 0.02, 5–90°, respectively, and a voltage of 45 kV was used. By means of field-emission scanning electron microscopy, the thicknesses of the deposited thin films were measured (Model: Gemini 500). By using an Al Kα X-ray analyzer in conjunction with XPS (Model: K-ALPHA+) with a spot size of 400 µm, excitation energy of 4.36 keV, energy step of 0.05 eV, and pass energy of 100 eV, the surface chemistries of the produced HEAs were determined. Argon etching was performed before characterization, and the C1 s peak was set to 284.6 eV.

## 3. Results and Discussion

After sputtering with varying ion densities, SEM was used to determine the coating thickness, as shown in Figure 2. Higher ion density (5 × 10^−5^ A/cm^2^) had more argon ions inside the sputtering chamber, resulting in thicker samples (the samples synthesized by this parameter are called thick hereinafter), and samples prepared at low ion density, i.e., 2 × 10^−5^ A/cm^2^, had less thickness with a difference of approximately 250 nm w.r.t thick coating, labeled as thin hereinafter.

For XRD characterization, the amorphous peak of the thin films was absorbed because, according to Equation (3), the atomic difference of Co_30_Cr_20_Ni_20_Mo_20_Ti_10_ was 5.57%, which is in the amorphous region for HEAs, as depicted in Figure 3 [36].

The heat treatment temperature range was 300–1000 °C for the HEA without copper. In this temperature range, the compositional difference between the dendritic ends and the center was eliminated, except in the case of the HEAs with copper in them because, in this temperature range, copper tended to segregate [37]. So, the prepared HEAs were heat treated within given temperature ranges.

The lattice distortion effect of the HEA with varying atomic sizes led to a significant loss of crystallization, which simplified its XRD patterns at high temperatures and atypically lowered the peak heights of the (310) and (200) planes at 600 °C and 800 °C, respectively [36].

The XPS results of the alloy are depicted in Figure 4, along with the corresponding Gaussian peak fitting results. The XPS results were obtained using a sample that was not heat treated (Table 2) [38,39,40,41,42]. Figure 4a depicts the XPS survey spectrum, which confirms the coexistence of all deposited elements in the Co_30_Cr_20_Ni_20_Mo_20_Ti_10_ HEA. By means of Ar etching for 17 min before the XPS examination, we decreased the heights of the O1s and C1s peaks considerably. Figure 4b depicts the XPS peaks of Cr at 574 eV and 583.5 eV, which correspond to Cr 2p_3/2_ and Cr 2p_1/2,_ respectively [41,43]. These peaks were further deconvoluted into two binding energies. In Figure 4c, cobalt has two prominent peaks at 778 and 793 eV, which correlate to Co2p_3/2_ and Co2p_1/2_, respectively. These intensities are further split into two regions in each peak [39,44]. The peak at 230.8 eV corresponds to (Mo3d_3/2_) Mo^+4^ [45] in Figure 4d, and in the same proximity, the peak centered at 227.5 eV corresponds to Mo3d_5/2_ [38,40].

The nickel peaks in Figure 4e were discretized into Ni2p_3/2_ peaks with spin-orbital characteristics at 852 and 859 eV and an Ni2P_1/2_ peak centered at 870 eV [40,44]. Similarly in Figure 4f, the Ti peaks are split into a Ti2P_3/2_ peak centered at 455 eV and two Ti2P_1/2_ peaks centered at 461 eV [42].

The results of the XPS studies confirmed that Mo^6+^ (Mo 3d_5/2_) and Mo^4+^ (Mo 3d_3/2_) were present in the synthesized coatings, and these ions were an integral part of the passive Cr_2_O_3_ films, in which Mo 3d_5/2_ controlled the film stability. MoO_4_^2−^ and MoO_2_ in these films, which had binding energies of around 227 eV and 230 eV, respectively, as listed in Table 1, acted as healers of any defects in the passive films [34].

For corrosion analysis, the open-circuit potential was stabilized for 3 h, and polarization curves were drawn between −0.5 V and 1.5 V at a scan rate of 10 mV/min on titanium (CP-II) substrate. The reference electrode for the electrochemical process was Ag/AgCl with 3M NaCl/saturated AgCl filling solution, and the electrode potential of the Ag/AgCl electrode is 0.02 V against the standard calomel electrode (SCE) [46]. The electrochemical impedance spectroscopy (EIS) experiment was run before drawing the Tafel plots in Figure 5 and Figure 6 because EIS is a non-destructive analysis. Moreover, the electrochemical properties of the synthesized HEA are summarized in Table 2.

As a biomaterial, the synthesized HEA surface coating must be characterized electrochemically to understand its behavior on implants from the perspective of avoiding inflammation through the release of ions into the human body. This is because the synergic effect of HEA and its corrosion characteristics vary significantly from those of known electrochemical processes.

The pitting corrosion on the coating surface is considered a major disadvantage for biomedical implants. In the discussion of the corrosion, we can add the fact that in the trans-passive region of the potentiodynamic curve, potential considerably greater than the corrosion potential pitting occurs (ref: ASTM-F746-0), as illustrated in Figure 5 and Figure 6.

The uncoated sample was a substrate of commercially pure titanium grade II substrate. The polarization curves represented in Figure 5 and Figure 6 were measured against the open-circuit potential. Ag/AgCl, and a Pt mesh were used as a reference and counter electrodes, respectively. Table 2 lists the current densities, corrosion resistance, and other important values obtained from the Tafel plots after fitting in EC-Lab software.

The atomic arrangement of synthesized HEA was amorphous, and high temperature leads to thermally active movement of atoms resulting in the formation of a small-ordered crystal structure, as depicted in Figure 3. Due to this movement, the compactibility of the coating is compromised, resulting in high current densities at elevated temperatures [47], as shown in Table 2. In the synthesized thin film, Mo-O and Cr-O generate a passivation mechanism by providing a healing effect to the coating. Therefore the amounts of molybdenum and chromium in the coating gives a passivation mechanism by the Cr-rich sigma (σ) phase [16]. This is corroborated by the high corrosion resistance of 8.69 × 10^9^ Ω of the thin film [34]. Cr (VI) is released from an alloy. It is only present for a very limited time because it is quickly reduced to the trivalent state in vivo [48].

Electrochemical impedance is a measurement of the resistance of coating for a given frequency and phase of alternating current. The higher the potential of a material, the lower its electron density at that potential, which means that there is more space for electrons to flow, and the surface is therefore resistant to oxidation. Electrochemical impedance characterization was performed by analyzing the Bode plots of the samples, as depicted in Figure 7. The frequency range used for the EIS measurements was 10 kHz to 1 mHz, and 1 sinusoidal wave was composed of 10 nodes of data points. Constant phase elements (CPEs) were determined using the following equation.
(6)$$Z_{CPE}\left(\omega \right)=\omega \frac{1}{{\left(i\omega \right)}^{\alpha }Q}$$
where *ω* is the angular frequency 0 ≤ *α* ≤ 1. At higher and lower frequencies, the circuit acted as capacitive and resistive, respectively.

The Bode plots show the variation of the impedance modulus and the phase angle of the CPEs in the film at the applied frequency of the sinusoidal signal. HEAs have higher impedance owing to the presence of metals that generate more ions in the resistive region, that is, at low frequencies ranging from 10^−2^ to 10^−1^, the thick and thin 400 °C samples show less contribution of charge transfer because their magnitudes are greater than 10^2^ to 10^3^ Ω on a logarithmic scale.

MTT dye was analyzed using a wavelength of 570 nm to perform a cell viability analysis. Osteoblast 3T3 cells were grown on processed Co_30_Cr_20_Ni_20_Mo_20_Ti_10_ HEAs, and their compatibility is shown in Figure 8. MTT analysis was performed at a significance level of 0.05 and a mean square value of 0.16. The sample heat treated at 600 °C indicated a significant difference in variability (S = 0.05). This HEA sample contained 10% Ti in three structural forms, namely alpha, beta, and alpha-beta phases, depending on the heat treatment condition and alloying additions. At 600 °C, the Ti was mostly α-Ti, leading to reduced cell viability [49]. Mo and Cr also functioned as beta stabilizers, for instance, in ASTM F 2066. Moreover, the biocompatibility of the samples decreased as their heat treatment temperature increased, as the electrical current started to flow within the cells [50]. The samples heat treated at 800 °C exhibited little crystallinity in the XRD (Figure 3), and accordingly, their biocompatibility was poor.

## 4. Conclusions

In this work, the thin HEA coating of 419 nm produced by sputtering at 2 × 10^−5^ A/cm^2^ ion density and without heat treatment outperformed all the other samples in terms of corrosion resistance of 8.69 × 10^6^ Ω-cm^2^ (Table 2) and biological cell viability (Figure 8). In our analysis, we studied an HEA by considering two variables, namely coating thickness in nanoscale and heat treatment temperature. In our analysis of the EIS property of the deposited HEA thin films, the WHT thin film formed a resistive circuit that protected biological cells and allowed current flow in the extracellular region. Therefore, this coating must have had a compacted passive layer. If the chromium content of the HEA exceeded 12%, molybdenum eliminated hydroxide ions from chromium. Similarly, nickel and cobalt enhanced the strength of the passivation layer, which increased its resistance to pitting corrosion [51], as shown in Table 2 [34]. The samples coated with Co_30_Cr_20_Ni_20_Mo_20_Ti_10_ at low ion density and without heat treatment produced relatively good osteoblast cell viability and corrosion resistance, as illustrated in Figure 9. This finding is solely attributable to the fact that coated samples tend to have less structure distortion than the other samples, and atomic mobility is noticeable in the XRD results from the remaining samples.

## Figures and Tables

**Figure 1 nanomaterials-13-01123-f001:**
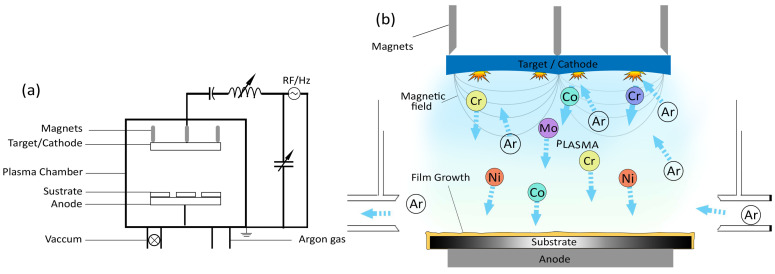
(**a**) Line diagram of radiofrequency (RF) magnetron sputtering; (**b**) Inside view of HEA sputtering process in the chamber.

**Figure 2 nanomaterials-13-01123-f002:**
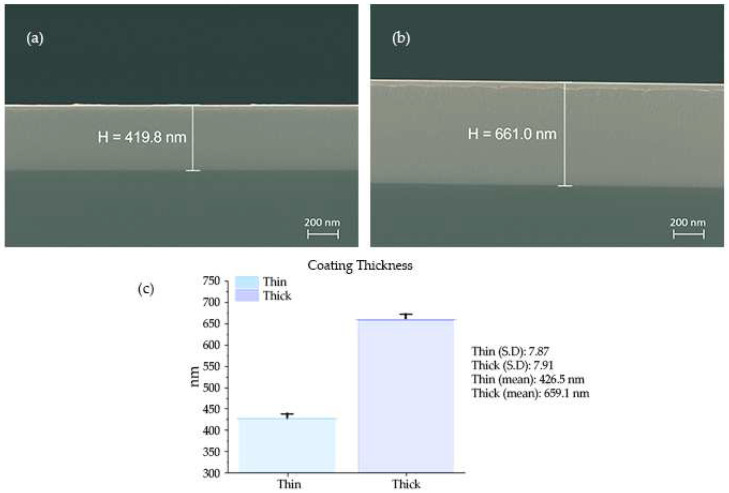
SEM analysis of HEA thin films synthesized at two different ion densities of (**a**) 20 µAcm^−2^ (thin film) and (**b**) 50 µAcm^−2^ (thick film). (**c**) Statistical differences between these two coatings.

**Figure 3 nanomaterials-13-01123-f003:**
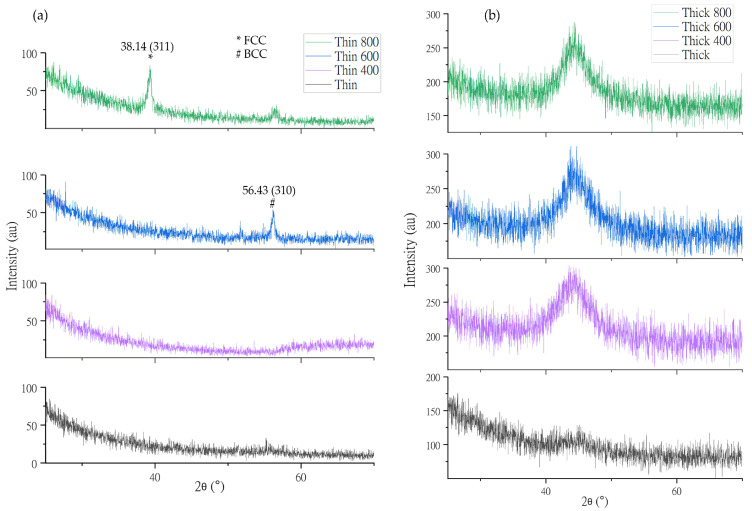
Amorphous XRD peaks of (**a**) thin and (**b**) thick HEA films without heat treatment (WHT) and heat treated at 400 °C, 600 °C, and 800 °C, respectively.

**Figure 4 nanomaterials-13-01123-f004:**
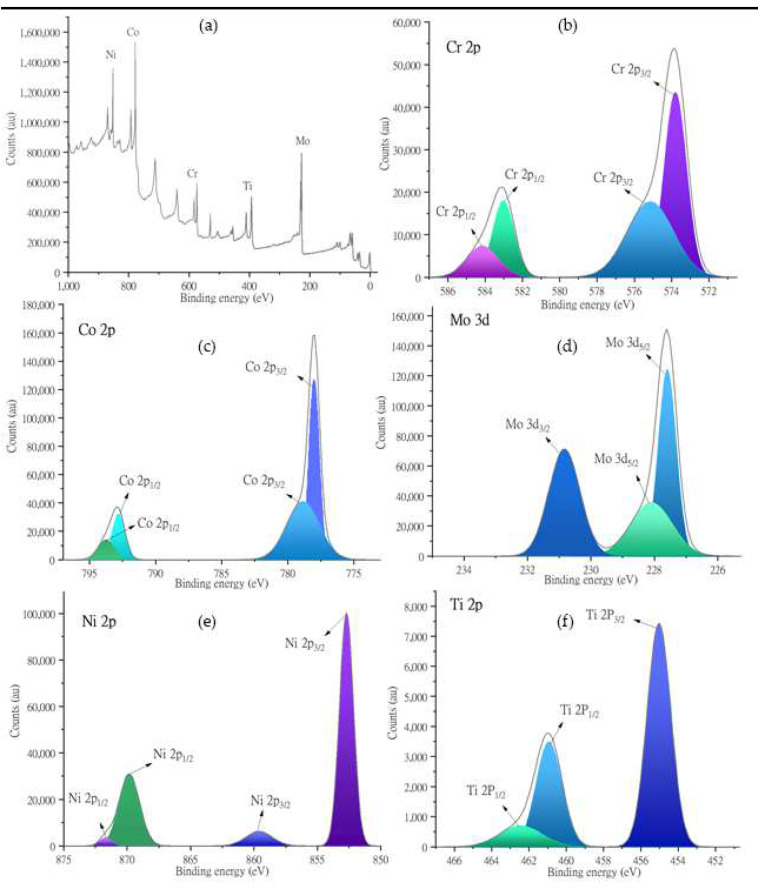
(**a**) XPS analysis of Co30Cr20Ni20Mo20Ti10 HEA-synthesized through magnetron sputtering and the corresponding peak fitting of (**b**) chromium, (**c**) cobalt, (**d**) molybdenum, (**e**) nickel, and (**f**) titanium.

**Figure 5 nanomaterials-13-01123-f005:**
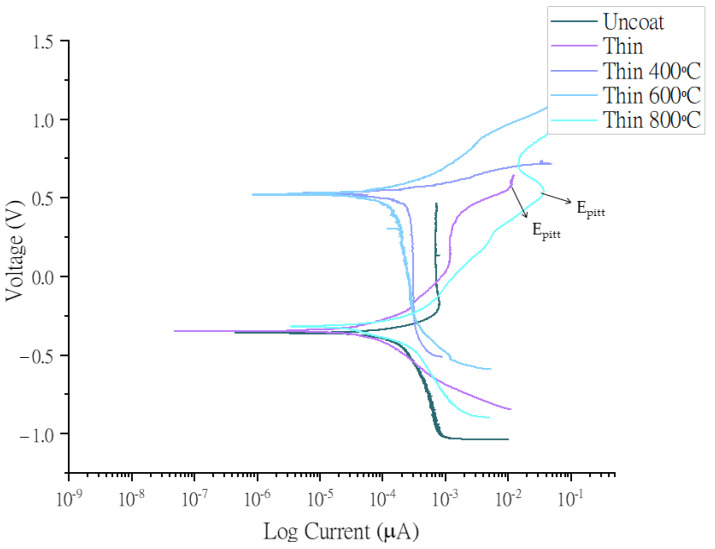
Polarization curves of Co30Cr20Ni20Mo20Ti10 alloy versus Ag/AgCl electrode in Hank’s solution. Working electrode was deposited at 2 × 105 ion density (thin coat) without heat treatment and with heat treatment at 400 °C, 600 °C, and 800 °C.

**Figure 6 nanomaterials-13-01123-f006:**
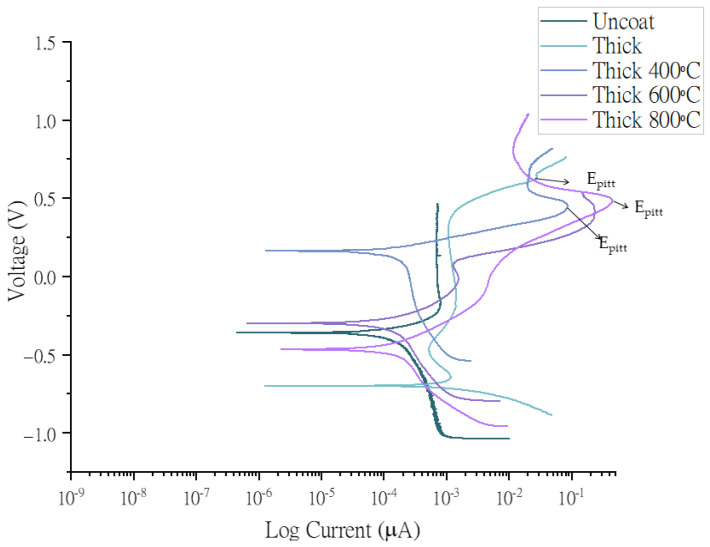
Polarization curves of Co30Cr20Ni20Mo20Ti10 alloy versus Ag/AgCl electrode in Hank’s solution. Working electrode was deposited at 5 × 105 ion density (thin coat) without heat treatment and with heat treatment at 400 °C, 600 °C, and 800 °C.

**Figure 7 nanomaterials-13-01123-f007:**
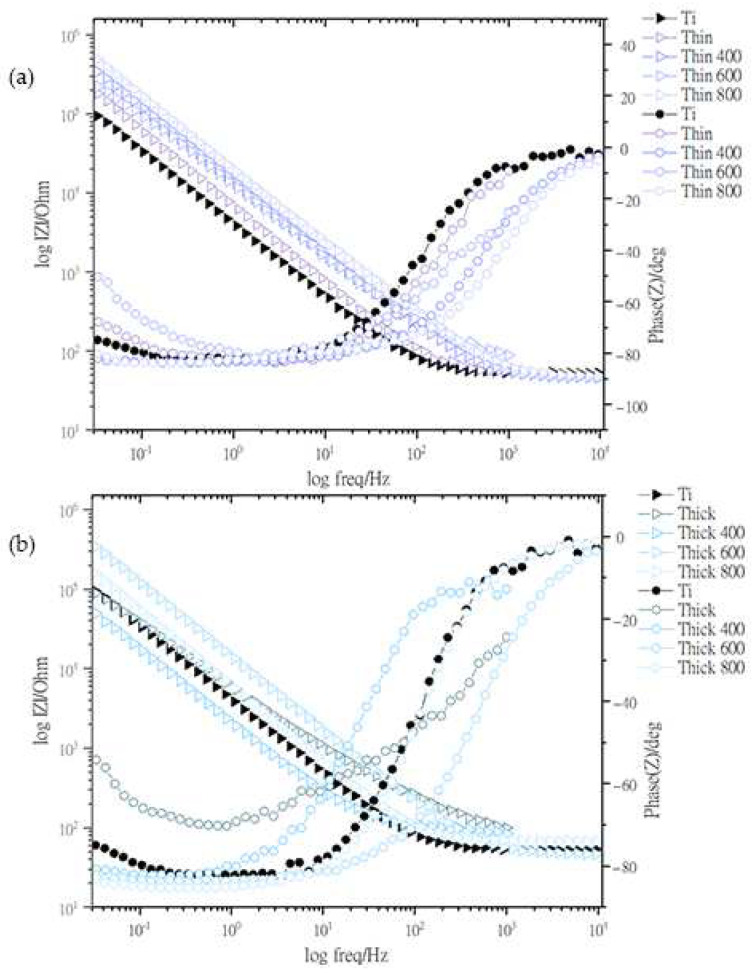
Impedance plots (Bode curve) of Co_30_Cr_20_Ni_20_Mo_20_Ti_10_ alloy versus Ag/AgCl electrode in Hank’s solution. Working electrode was deposited at (**a**) 2 × 10^5^ ion density (thin coat) without heat treatment and with heat treatment at 400 °C, 600 °C, and 800 °C (**b**); (**a**) 5 × 10^5^ ion density (thin coat) without heat treatment and with heat treatment at 400 °C, 600 °C, and 800 °C.

**Figure 8 nanomaterials-13-01123-f008:**
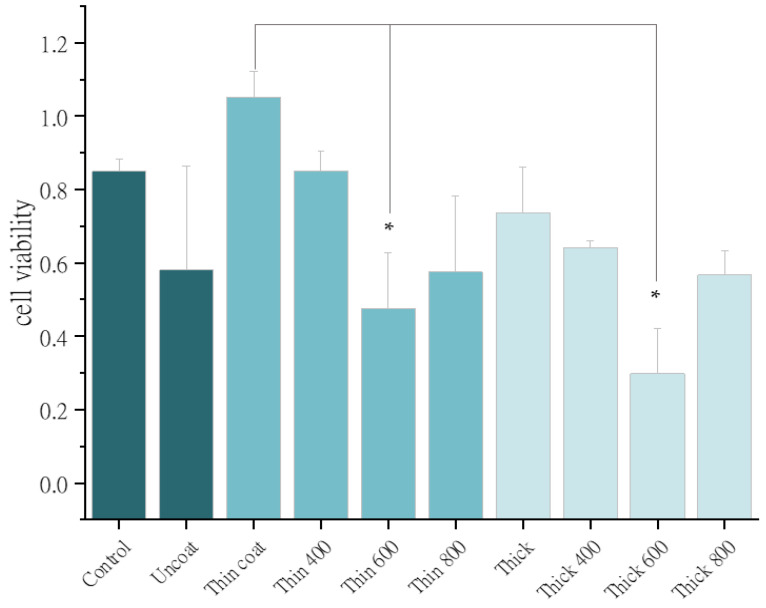
MTT analysis of uncoated titanium surface, coated Co_30_Cr_20_Ni_20_Mo_20_, and heat treated surfaces. The significant difference between thin coat and samples heat treated at 600 °C is marked with hysteric.

**Figure 9 nanomaterials-13-01123-f009:**
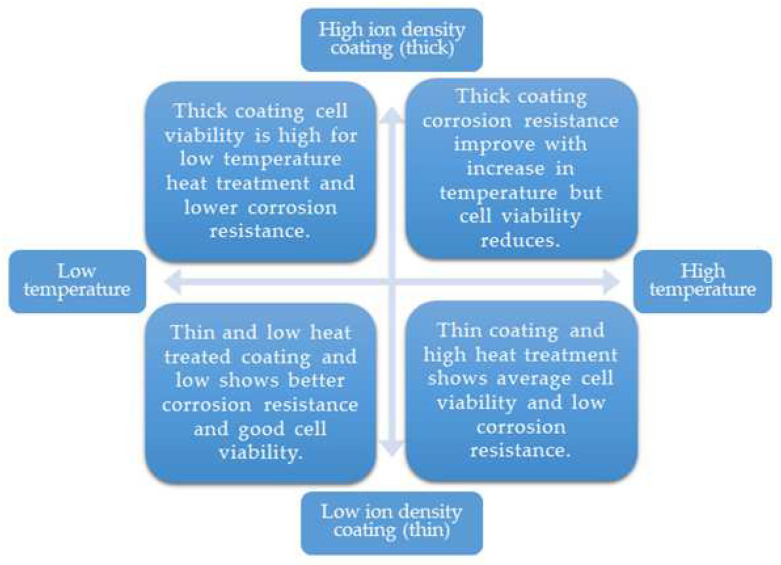
Effect of heat treatment and coating thickness on electrochemical property and osteoblastic cell viability.

**Table 1 nanomaterials-13-01123-t001:** Elemental composition and orbital configuration of deposited Co_30_Cr_20_Ni_20_Mo_20_Ti_10_ HEA.

Element	Atomic Percentage (%)	Binding Energy (eV)	FWHM (ΔeV)	Chi-Square (×10^6^)
Cobalt	31.89	777~794		3.25
Co 2p_3/2_	80.41 of 31.89	778.42	1.10	
Co 2p_1/2_	19.59 of 31.89	793.29	1.52	
Chromium	17.42	573~585		5.74
Cr 2p_3/2_	74.20 of 17.42	574.46	1.77	
Cr 2p_1/2_	25.80 of 17.42	583.58	1.75	
Molybdenum	21.93	227~231		3.65
Mo 3d_5/2_	62.60 of 21.93	227.59	0.79	
Mo 3d_3/2_	37.40 of 21.93	229.46	1.16	
Nickel	20.51	869~852		4.08
Ni 2p_3/2_	62.24 of 20.51	852.70	1.29	
Ni 2p_3/2_	7.29 of 20.51	859.60	2.46	
Ni 2p_1/2_	30.47 of 20.51	871.69	2.01	
Titanium	8.24	455~462		0.34
Ti 2p_3/2_	59.52 of 8.24	455.02	1.50	
Ti 2p_1/2_	40.48 of 8.24	461.67	1.83	

**Table 2 nanomaterials-13-01123-t002:** Electrochemical properties of HEA with reference to uncoated titanium CP-II.

Property	Uncoated	Thin Coat	Thin Coat 400	Thin Coat 600	Thin Coat 800	Thick Coat	Thick Coat 400	Thick Coat 600	Thick Coat 800
E_corr_ (mV)	−363.08	−347.46	527.04	522.49	−318.95	−698.23	163.09	−297.67	−463.64
I_corr_ (µA)	0.313	0.040	0.111	0.122	0.108	1.851	0.145	0.126	0.088
Corrosion rate (mmpy)	0.0108	0.0014	0.0038	0.0042	0.0037	0.0641	0.0050	0.0043	0.0030
Resistance (Ω)	1.16 × 10^6^	8.69 × 10^6^	4.74 × 10^6^	4.28 × 10^6^	2.96 × 10^6^	3.77 × 10^5^	1.12 × 10^6^	2.36 × 10^6^	5.26 × 10^6^
E_pit_ (V)	--	0.569			0.575	0.629	0.434		0.486
Log I_pass_ (mA)	--	0.010			0.033	0.029	0.080		0.433

## Data Availability

Data will be available upon request.

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
