# Peer review of "The Surface Properties of Implant Materials by Deposition of High-Entropy Alloys (HEAs)"

_nanomaterials, 2023, doi:10.3390/nano13061123_

Round 1

Reviewer 1 Report

In its current form, the paper contains a great deal of weaknesses, ambiguities and insayings. Besides, the English language of this work is particularly poor. In the present version, the manuscript is not suitable for publication. Below are several examples of banning comments, but one can mention much more of similar deficiences.

1. Average atomic radius of 164.2 nm (!) - this is the same order of magnitude as the thickness of the coatings (Line 111)

2. What (specifically) was the concentration of O2 in the aggressive solution (L.120,121)? Which way was the limiting current of O2 diffusion calculated or determined?

3. What was the substrate for the HEA layers? The info that it was titanium is mentioned only at the final parts of the work.

4. How was compactness and continuity of the HEA layer determined? As it is well known, any discontinuities (open porosity) across this type of layer lead to violent galvanic effects (accelerating corrosion)

5. The potential scan rate was very fast. How do the Authors know that only Faradaic processes are recorded on the polarization curves?

6. Figure captions must be completed without reference to the text (e.g. versus which reference electrode was the electrode potential measured/expressed? Which was the working solution, etc? – Figs 5-7)

7. What concentration of chloride ions was applied (assumed?) in the the Ag/AgCl reference electrode?  Which was the value of equilibrium potential of this electrode?

8. Why are the polarization curves for the substrate material „finished” at E = 0.5V (that is, much earlier than for HEA coatings)? Does the pure titanium also undergo pitting corrosion or not?

9. In the methodology and in figure captions (Figs 5-7) it must be elearly indicated what was the working solution.

10. Why the Authors discuss the segregation of Cu in the coatings? (After all, the tested HEAs did not contain copper) - L.160-161

11. Many parts of the text are incomprehensible (e.g. L.208-210)

12. Corrosion parameters are calculated by software and not subjected to any critical discussion. As it is known from theory of corrosion, the corrosion current density of passivating metal should be equal to density of current within passive region. Why these two rates differ by each other (Table 2)? For Icorr = 0.1 uA (or 0.0001 mA), the log(Ipass) values should be negative (-1 and -4, respectively). By the way, it is a misunderstanding to provide Ecorr values with the accuracy of 5 significant digits (figures).

13. The appearance of pitting corrosion is disqualifying for the HEA materials of implants - the Authors seem to ignore this fact.

Author Response

Hello respected reviewer,
I have spent a week studying and reading the comments. All that points are valid and had been corrected in attached file.
Further suggestions and comments will handle with care.
Kind regards.

Reviewer 2 Report

 The  manuscript title and abstract are suitable and according to paper content.  It is well written with an appropiate research design for  elaboration of a HEA coating on Ti implant material, and its surface and electrochemical properties. Being  thin films elaborated by means of high vacum radio frequency  magnetron (HVRF) sputtering, their thinckness are in the nanolevel domain, and Nanomaterial journal could be a  choice, despite the fact that section 2d and Carbon Nanomaterials is not the best selection.

Introduction and subchapter References present the paper aim and novelty, but there are things to be improved such as

1toxicological risks of the cobalt–chromium alloys used in  implant materials are not dicussed despite the fact that  are a part of new EU Medical Devices Regulation (MDR) (2017/745) which was supposed to  be applied in May 2021 [This act has been changed. Current consolidated version: 24/04/2020]. With the new regulation, Co metal will be considered  carcinogenic, mutagenic, and a substance toxic to reproduction.                    2 it is  a need for an extended data treatment and I do propose the evaluation of porosity from polarization data to be correlated with  films thickness                                                                                                         3 the absence  of  peaks values for XRD in the figure                                   4 a statistical treatment of data it will be in the manuscript benefit

Author Response

Hello respected reviewer,
I have spent a week studying and reading the comments. All that points are valid and had been corrected in attached file.
New figures and tables will be added in the manuscript, once it approved.
Further suggestions and comments will handle with care.
Kind regards.

Round 2

Reviewer 2 Report

I do consider that the authors answer is partially complete. The electrochemical stability experiments need to have a more complete way of data treatment

Author Response

Hello respected reviewer,
The samples coated with Co30Cr20Ni20Mo20Ti10 at low ion density and without heat treatment produced relative good osteoblast cell viability and corrosion resistance, as illustrated in attached file. This finding is solely attributable to the fact that coated samples tend to have less structure distortion than the other of the samples, and atomic mobility is noticeable in the XRD results from the remaining samples.

Add a new conclusive figure to describe coating thickness as a function of electrochemical behavior and cell viability.

Kind regards.
